# Chiral donor–acceptor azetines as powerful reactants for synthesis of amino acid derivatives

Kostiantyn O. Marichev [1], Kuiyong Dong[1], Lynée A. Massey [1], Yongming Deng[1], Luca De Angelis[1], Kan Wang[1], Hadi Arman[1] & Michael P. Doyle[1]*

Coupling reactions of amines and alcohols are of central importance for applications in chemistry and biology. These transformations typically involve the use of a reagent, activated as an electrophile, onto which nucleophile coupling results in the formation of a carbon-nitrogen or a carbon–oxygen bond. Several promising reagents and procedures have been developed to achieve these bond forming processes in high yields with excellent stereo-control, but few offer direct coupling without the intervention of a catalyst. Herein, we report the synthesis of chiral donor–acceptor azetines by highly enantioselective [3 + 1]-cycload-dition of enoldiazoacetates with aza-ylides and their selective coupling with nitrogen and oxygen nucleophiles via 3-azetidinones to form amino acid derivatives, including those of peptides and natural products. The overall process is general for a broad spectrum of nucleophiles, has a high degree of electronic and steric selectivity, and retains the enantio-purity of the original azetine.

[1] Department of Chemistry, The University of Texas at San Antonio, San Antonio, TX 78249, USA. *email: michael.doyle@utsa.edu

rreversible ring opening of the strained 2-azetidinone four-membered ring, which is one of the key biomolecular events during both the antibiotic action of β-lactams and their inhibition by β-lactamases[1], is a model for nucleophile coupling. Because of their chemically controlled ring opening, 2-azetidinones are widely used for the synthesis of heterocycles, β-amino acids, and their derivatives[2–4]. 3-Azetidinones, by contrast, are less well established[5] even though they have the potential for analogous nucleophilic carbonyl-carbon cleavage to form amine derivatives (Fig. 1a) if an activating electron-withdrawing group (EWG) is located at the 2-position; but the key to realizing this potential lies in the design of a 3-azetidinone capable of nucleophile coupling.

A classic approach to nucleophile coupling is the retro-Claisen reaction of β-ketoesters[6] that would require the construction of 2-carboxylate substituted 3-azetidinones, but the basic methods available for their ring-closing formation are the same as those desired for their ring-opening which is favored by ring strain[7–10]. Although one example of N-Cbz protected 2-carboxylate substituted 3-azetidinones is reported, its synthetic utility has not been pursued due to its inherent instability to nucleophilic ring opening[11]. Alternative methodologies proceeding to 2-azetine-2-carboxylate structures applied to the formation of the 3-azedidinone analogues, either through [2 + 2]-cycloaddition[12], from 3-substituted 2-azetines (lithiation)[13–17], or with N-Boc-3-azetidinone (coupling reactions)[18], have not been suitable for 2-carboxylate derivatives. In addition, attempted copper(I)-catalyzed [3 + 1] cycloaddition of alkenyldiazoacetates and iminoiodinanes to form the requisite 3-azetidinones has also been unsuccessful[19]. To circumvent these approaches, we present the formation of an enolate precursor to 2-carboxylate substituted 3-azetidinones that appears to be available by [3 + 1]-cycloaddition (Fig. 1b).

Previous research from our laboratory has established that [3 + 1]-cycloaddition of silyl group protected enoldiazoacetates with α-acyl sulfur ylides is effective in forming donor–acceptor cyclobutene derivatives in good yields and high stereocontrol[21]. With a corresponding 2-azetine derivative obtained in good yield and high enantioselectivity, desilylation, forming the desired 3-keto-2-carboxylate, could provide the chemical framework for nucleophile coupling that would attach the amino acid framework of the 3-azetidinone to the nucleophile.

In order to achieve strain-induced ring opening of 3-azetidinones, we present here the synthesis of highly optically enriched donor–acceptor azetines and their facile ring opening in high yield with a broad spectrum of nucleophiles under mild conditions. Highly enantioselective [3 + 1]-cycloaddition of silyl-protected enoldiazoacetates occurs with aza-ylides using chiral copper (I) catalysis. The 2-azetidine cycloaddition products generate 3-azetidinones by reactions with nucleophiles that produce a broad spectrum of amino acid derivatives with high efficacy and complete retention of enantiopurity. This retro-Claisen reaction aided by strain release uncovers a methodology for the attachment of chiral amide, peptide and ester units to a variety of amines and alcohols, and tolerates a broad scope of nucleophiles, including naturally occurring amines, alcohols, amino acids, and other nitrogen-based nucleophiles. Mechanistic studies confirm initial desilylation followed by nucleophilic retro-Claisen ring opening. The mild reaction conditions, high enantiocontrol, broad scope of nucleophiles for the ring opening of donor–acceptor azetines, and ability to perform the reaction in aqueous media demonstrated in this work portray a process that will have wide applications.

## Results

### Development of the [3 + 1]-cycloaddition reaction.
Application of N-acylimido sulfur ylides[21–23] and enoldiazoacetates to the same

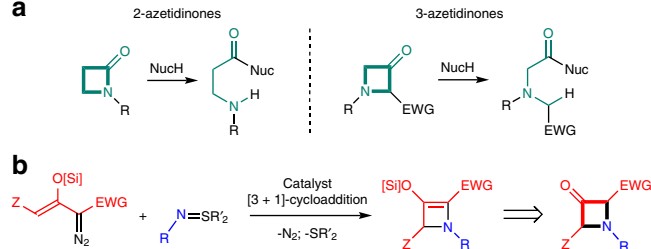

**Fig. 1** Strain-induced ring opening of azetidinones and [3 + 1]-cycloaddition. **a** Amide ring opening of 2-azetidinones compared with retro-Claisen ring opening of EWG-activated 3-azetidinones. **b** Generation of 3-azetidinones via catalytic [3 + 1]-cycloaddition of silyl-protected enoldiazoacetates with imido sulfur ylides.

catalysts and conditions that were successful with their carbon analogues was unsuccessful even at elevated temperatures due to a lack of reactivity of the N-acylimido ylide. Use of N-arylimido sulfur ylides (S,S-disubstituted N-arylsulfilimines)[24–26], however, allowed cycloaddition to proceed smoothly at room temperature. As previously described for the corresponding [3 + 1]-cycloaddition that formed donor–acceptor cyclobutene derivatives[21], only copper(I) catalysis was effective for this transformation; and Cu(MeCN)₄PF₆ was the catalyst of choice in the formation of 2-azetines. Product yields were the highest in dichloromethane, and diphenylsulfur ylides gave higher product yields than their dimethyl or methylphenyl analogues (see Supplementary Table 1). Reactions were performed at room temperature to avoid electrocyclic ring opening of the azetine[27–29]. [3 + 1]-Cycloaddition occurred with the triisopropylsilyl(TIPS)-protected enoldiazoacetate, but not with the tert-butyldimethylsilyl(TBS)-protected enoldiazoacetate. With these optimizations methyl N-(p-chlorophenyl)-3-OTIPS-2-azetine-2-carboxylate **3** was formed in 80% isolated yield (Fig. 2a).

To introduce chirality into the 2-azetine-2-carboxylate, a substituent at the terminal vinyl position of enoldiazoacetate **1** is required. Previous reports on enoldiazoacetates described the synthesis and uses of only two TBS- and TIPS-protected enoldiazoacetates having terminal vinyl substituents (4-Me and 4-Ar)[20,30–36], and both of their geometrical isomers were formed in the case of TIPS-derivatives. We have provided a synthetic solution to this challenge that allows dominant formation of the Z-isomer (Z:E = > 20:1) for these substituted enoldiazoacetates[37] and only the Z-isomer undergoes [3 + 1]-cycloaddition[39]. To effect asymmetric induction for 2-azetine ring formation, we initially selected methyl (Z)-3-OTIPS-2-diazo-3-pentenoate **1b** with N-(p-chlorophenyl)imido diphenylsulfur ylide **2a** and performed the cycloaddition reaction under the optimized conditions with catalysis by Cu(MeCN)₄PF₆ coordinated to chiral side-arm bisoxazoline (sabox) ligand **L1** (Fig. 2b). The use of ligand **L1** resulted in the highest yield and enantioselectivity (71% yield, 75% ee).

Although the yield and enantioselectivity for **3b** obtained with **L1** were only moderate, we proceeded to vary substituents at the 4-position of enoldiazoacetate **1** in order to determine if these substituents influence product formation and selectivity. A general procedure was established for the introduction of substituents to the 4-position of enoldiazoacetate **1**[38]; and, using **2a** as the optimum sulfilimine, [3 + 1]-cycloaddition was performed under optimum conditions. The initial reaction of **1b** (Z:E = 3:1) with a 50% molar excess of **2a** showed complete loss of Z-**1b**, but retention of E-**1b** and a 75% ee for **3b** (Fig. 2b). This observation prompted us to use an excess of the 4-substituted enoldiazoacetate over sulfilimine **2a** to reflect the actual stoichiometric amount of

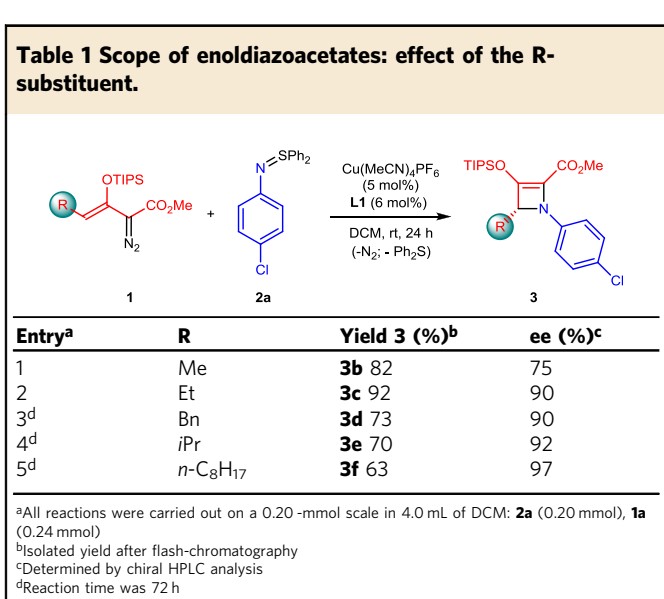

**Fig. 2** Azetine ring construction. **a** [3 + 1]-Cycloaddition of methyl 3-triisopropylsil(TIPS)oxy-2-diazo-3-butenoate (**1a**) with *S,S*-diphenyl *N*-(*p*-chlorophenyl)sulfinylimine (**2a**) catalyzed by copper(I) tetrakis(acetonitrile). **b** Asymmetric synthesis of donor–acceptor azetine using copper(I) catalysis with chiral sabox ligand **L1**.

**Table 1 Scope of enoldiazoacetates: effect of the R-substituent.**

| Entry[a] | R | Yield 3 (%)[b] | ee (%)[c] |
|---|---|---|---|
| 1 | Me | **3b** 82 | 75 |
| 2 | Et | **3c** 92 | 90 |
| 3[d] | Bn | **3d** 73 | 90 |
| 4[d] | *i*Pr | **3e** 70 | 92 |
| 5[d] | *n*-C$_8$H$_{17}$ | **3f** 63 | 97 |

[a]All reactions were carried out on a 0.20 -mmol scale in 4.0 mL of DCM: **2a** (0.20 mmol), **1a** (0.24 mmol)
[b]Isolated yield after flash-chromatography
[c]Determined by chiral HPLC analysis
[d]Reaction time was 72 h

**Table 2 Scope of enoldiazoacetates: effect of the carboxylate group.**

| Entry[a] | R$^1$ | R$^2$ | Yield 3 (%)[b] | ee (%)[c] |
|---|---|---|---|---|
| 1 | Et | *i*Pr | **3g** 82 | 89 |
| 2 | Et | Bn | **3h** 87 | 92 |
| 3 | Et | 4-BrBn | **3i** 90 | 90 |
| 4 | Et | 4-OMeBn | **3j** 95 | 99 |
| 5[d] | Et | 3,4,5-triOMeBn | **3k** 70 | 98 |
| 6 | Et | 4-CF$_3$Bn | **3l** 73 | 87 |
| 7 | Me | 4-OMeBn | **3m** 77 | 88 |

[a]All reactions were carried out on a 0.20 mmol scale in 4.0 mL DCM: **2a** (0.20 mmol), **1a** (0.24 mmol)
[b]Isolated yield after flash chromatography
[c]Determined by chiral HPLC analysis
[d]Reaction time was 48 h

the *Z*-isomer in the **Z-1**/**E-1** mixture. When the reaction of **1b** (*Z*:*E* = > 20:1) with **2a** was repeated using a (1.2):1 ratio **1b**/**2a** [vs. 1:(1.5) reported in Fig. 2b], this modification resulted in an increased yield of **3b** to 82% (entry 1, Table 1) with the same ee value of 75%. To our good fortune, changing the methyl substituent at the 4-position of **1** to ethyl not only improved the enantioselectivity for the [3 + 1]-cycloaddition to 90% ee but also resulted in an increase of the isolated yield (92%) of **3c** (entry 2, Table 1). Further elaboration of the substituent at the 4-position with benzyl (**3d**), isopropyl (**3e**), and *n*-octyl (**3f**) under the same conditions led to a modest decrease in reactivity, apparently due to steric effects, and lowered product yields, but % ee values were comparable with or higher than that of **3c** (90−97% ee).

To identify a possible further improvement in enantiocontrol, we investigated the influence of the carboxylate ether group (size and electronic effects) of enoldiazoacetates **1**. With an Et (R$^1$) substituent at the 4-position (Table 2) introduction of an isopropyl group as R$^2$ (**1g**) resulted in a decrease of azetine yield without a change in enantioselectivity (entry 1; Table 2). Notably, the corresponding *tert*-butyl enoldiazoacetate (R$^2$ = *t*Bu) resulted in only trace amounts of the [3 + 1]-cycloaddition product. Neither benzyl (**1h**) nor 4-bromobenzyl (**1i**) substituted enoldiazoacetates provided any noticeable improvement in

enantiocontrol (90−92% ee) and yields (87−90%) (entries 2,3; Table 2). Surprisingly, the *p*-methoxybenzyl (PMB) ester provided a remarkable level of enantiocontrol (99% ee) and also produced **3j** in 95% yield (entry 4; Table 2). A very similar ee value (98% ee) was obtained for the 3,4,5-trimethoxybenzyl derivative **3k**, however, the reaction time for this reaction was extended to 48 h in order to achieve full conversion (entry 5; Table 2). As expected, the presence of the electron-withdrawing CF$_3$ group at the 4-position of phenyl ring (**1l**) resulted in a decrease of both the yield (73%) and enantioselectivity (87% ee) of azetine **3l**. To determine that the effect of the PMB group as R$^2$ might be general, we prepared *p*-methoxybenzyl 3-OTIPS-2-diazo-3-pentenoate **1m** and performed the [3 + 1]-cycloaddition reaction (entry 7; Table 2): enantioselectivity was improved from 75% (**3b**, R$^2$ = Me) to 88% ee (**3m**, R$^2$ = PMB).

**Nucleophilic ring opening of donor–acceptor azetines**. That ring opening would be a facile process of these donor–acceptor azetines was not initially obvious. Five- and six-membered ring silyl-protected β-enolcarboxylates are well known to form β-ketoesters after desilylation[39–42]. However, when azetine **3b**

was treated with the classic TBAF to effect desilylation, a mixture of ring-opened products was obtained under typically mild conditions. This observation suggested that initial enolate formation had occurred and that subsequent nucleophilic reaction on the β-keto ester or its equivalent effected strain-induced ring opening. To determine the extent of nucleophilic ring opening with strain release of donor–acceptor azetines, we have treated them with a variety of nitrogen and oxygen nucleophiles. We assumed that TIPS group removal from 2-azetine-2-carboxylates **3** occurs under mild conditions to generate the 3-azetidinone carboxylate structure, which then undergoes ring opening with the excess of a nucleophile (Fig. 3). This concept of strain release through carbon–carbon σ-bond cleavage from 3-azetidinone carboxylates bond is uncovered in this work, and this nucleophile coupling opens doors to enormous opportunities in the synthesis of chiral amino acid derivatives and relevant substances of biological interest with high optical purity.

2-Azetine-2-carboxylates **3c** and **3j** were the substrates of choice in most cases because of their availability and optical purity (90 and 99% ee, respectively). Initial assessment of reactivity was carried out by reactions of **3c** and **3j** with 2.5 equiv. of benzylamine in DCM at room temperature (Table 3). Ring-opened products that contain one amide bond were formed within 48 h in excellent isolated yields (**4a–d**, **5**; 90–96%). Moreover, we observed the complete retention of optical purity in the products of the nucleophile coupling reactions (see Supplementary Figs. 159–170). The scope of reactions between 2-azetine-2-carboxylate **3** and amines tolerates substituents other than chloride in the aromatic ring of **3**. We did not see a substantial difference in either product yields or enantioselectivities with electron-withdrawing substituents on the benzene ring (**4a–c**). However, with the electron-donating methyl group at the meta position of the *N*-aryl group, the corresponding 2-azetine-2-carboxylate was obtained with diminished enantiopurity (74% ee) that was carried over to the ring-opened product **4d**. The ester functionality remained intact under these reaction conditions even when 4 equiv. of benzylamine was used. However, consistent with nucleophilic reactions that form ionic intermediates[43,44], changing the polarity of the solvent played a significant role in increasing the reactivity of **3c** toward benzylamine, so that with 4 equiv. of benzylamine or 4-bromobenzylamine in THF/water 1:1 (v/v) chiral diamide derivatives **6a,b** were formed in high yields (93 and 85%, respectively) within 12 h. This was an unexpected observation because the classic reaction of an ester with amines is very sluggish at room temperature. A control reaction of monoamide **4** with an excess of benzylamine (2 equiv.) resulted in a quantitative yield of diamide **6** in 12 h. Apparently, formation of the first amide unit activates the carbonyl group of the ester via intramolecular hydrogen bonding in water (polar protic solvent), and therefore favors the nucleophilic substitution by benzylamine on the ester group.

We investigated the regioselectivity of the ring-opening reaction of **3c** with *N*-cyclohexyl-1,3-propanediamine (primary-secondary diamine). Formation of the amide bond occurred exclusively at the primary amine position of the diamine, and the monoamination product **7** was isolated in 92% yield. Inspired by the results with benzylamine, the remarkable levels of regioselectivity with *N*-cyclohexyl-1,3-propanediamine, and the solvent effect on product formation, we examined reactions of 2-azetines **3c** and **3f** with biologically relevant amines and natural amino acids. Monoamide derivatives of tryptamine (**8** and **9**), benzyl-protected *L*-proline (**10**), and glutamic acid diethyl ester (**14**), for example, were obtained in 84–88% isolated yields in reactions carried out in dichloromethane. Ring opening of **3c** with a natural polyamine, spermine, in DCM occurred selectively at the terminal primary amine position of spermine but, unlike with other amines, formed chiral diamide **15**

as the major product (63% yield). The reaction of **3c** with *tert*-butylamine, a sterically hindered primary amine, occurred with similar efficacy as that with benzylamine (92% yield). Treatment of azetine **3c** with aromatic amines, which are weaker nucleophiles, showed negligible conversion in DCM at room temperature, but heating **3c** with aniline (3 equiv.) at 65 °C in 1,2-dichloroethane (DCE) for 24 h resulted in ring-opening nucleophilic coupling; however, **13a** was formed in only 65% yield together with the product from the known thermal electrocyclic ring opening of **3c**[27–29]. Use of electron-rich 4-(dimethylamino)aniline with **3c** in nitromethane at room temperature increased the yield of the ring-opened product to 93% (**13b**). The reaction of **3c** with (*R*)-phenylglycinol (2.5 equiv.) in DCM at room temperature was sluggish but highly selective toward the amino group, affording monoamidation product **11** (d.r. >20:1) in 90% yield after 4 days. We were also interested if *L*-lysine methyl ester (basic form) was able to provide high regioselectivity in the ring-opening reaction with **3c** carried out in DCM. Indeed, remarkable regiocontrol (at the terminal amino group) was achieved in the formation of monoamide derivative **16** (90% yield of a single diastereomer). The same regiocontrol was obtained in the reaction of 2-azetine carboxylate **3c** with *L*-lysine (4 equiv.) in THF/water 1:1 (v/v), but this reaction also resulted in hydrolysis of the ester to the carboxylic acid (**17**, 75% yield) under the reaction conditions. Reactions with *tert*-butylamine and pyrrolidine in THF/water 1:1 (v/v), unlike that with benzylamine, resulted in monoamidation and hydrolysis of the ester to form amidocarboxylic acids **18** (80%) and **19** (72%), rather than in the formation of a diamide.

As expected, the ring-opening reactions of 2-azetine-2-carboxylates with the weaker alcohol nucleophiles occurred at slower rates (Table 4). However, very high yields (up to 92%) of chiral diesters **20–22** were obtained with complete retention of enantiopurity from the reactions of **3c**, **3j**, and **3k** with methanol used as the solvent at 65 °C. Use of higher-molecular-weight primary alcohols resulted in a decrease of their reactivity with 2-azetine-2-carboxylates. The yield of diester **23** obtained with ethanol (66%) at 65 °C after 24 h was similar to that obtained from the reaction with aniline (65%), but a much larger excess of the nucleophile was used in this case (ethanol was used as the solvent). Geraniol (a naturally occurring primary alcohol)[45,46] and ethylene glycol also formed the corresponding diesters **24** and **25** in moderate yields at 65 °C after 24 h using only 4 equiv. of alcohol. In all reactions performed at elevated temperatures, the main competing reaction was electrocyclic ring opening of 2-azetine carboxylate **3c**[27–29].

Besides amines, amino acids, and alcohols, we have tested other relatively strong nitrogen-based nucleophiles and tetrabutylammonium fluoride (TBAF) (Table 5). The monomethyl ester of chiral dicarboxylic acid **26** was obtained in 70% yield by a simple treatment of 2-azetine-2-carboxylate **3c** with a THF solution of TBAF. Alternatively, compound **26** was obtained in near quantitative yield by treatment of azetine **3c** with 5 equiv. of water in nitromethane at room temperature in 12 h. The reactivity of **3c** with phenylhydrazine was higher than that with aniline, and chiral monohydrazide **27a** was obtained in 87% yield at 50 °C in DCE after 12 h together with minor amounts (<10%) of the electrocyclic ring-opening product from **3c**. The reaction of electron-deficient 4-nitrophenylhydrazine with **3c** in nitromethane at room temperature resulted in high yield (88%) of the ring-opened product **27b**. The use of aqueous solutions of hydroxylamine, hydrazine, and ammonia for the reaction with **3c** in THF led to the formation of two C–N bonds and afforded chiral dihydroxamic acid **28**, dihydrazide **29**, and diamide **30** in excellent yields (up to 96%) in 12 h. Efforts to perform selective reactions leaving the ester group intact were not successful. An

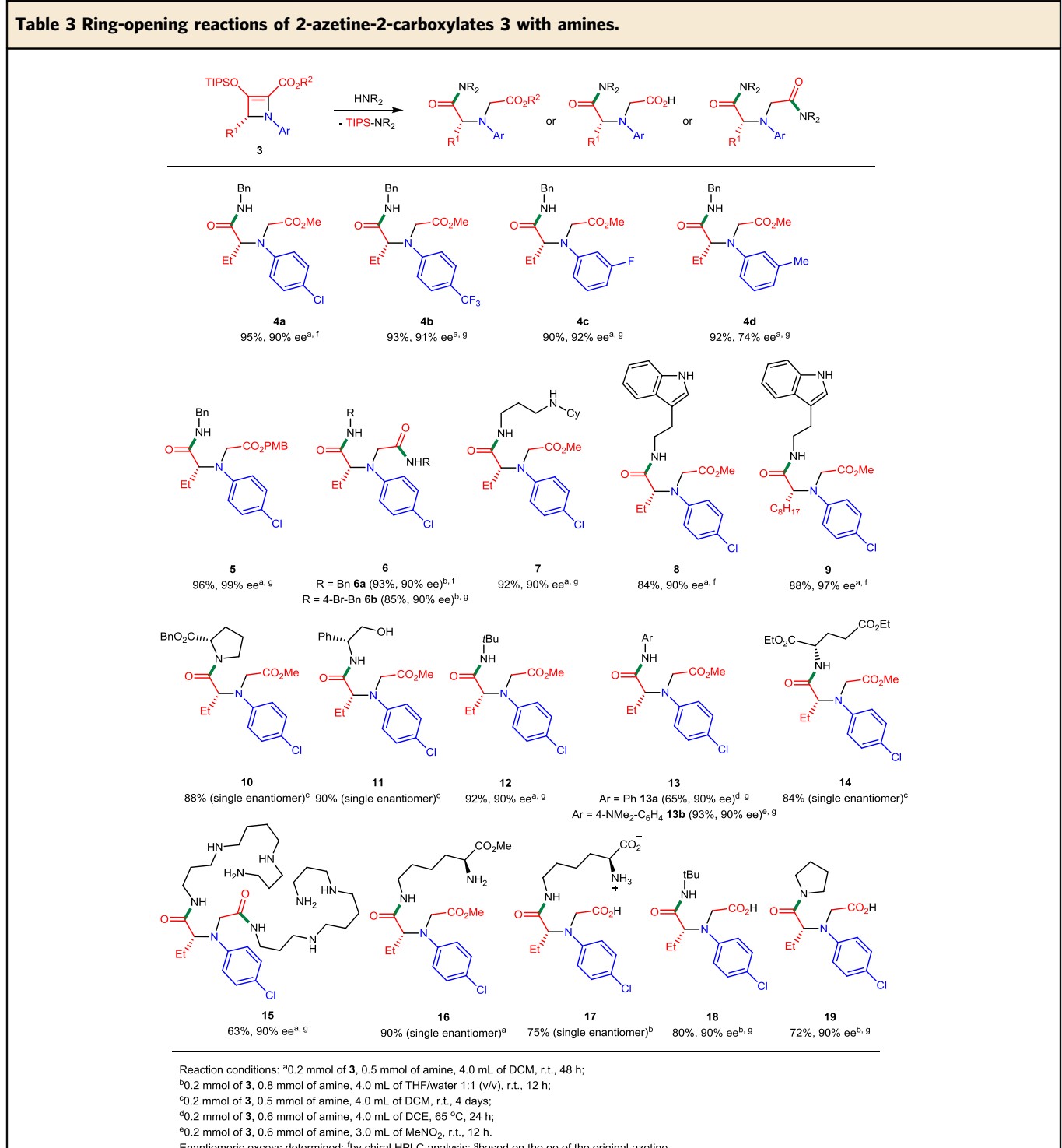

**Fig. 3** Strain-induced nucleophilic ring opening of donor–acceptor azetines. Desilylation of the azetine forms the activated 3-azetidinone that undergoes nucleophilic retro-Claisen ring opening.

**Table 3 Ring-opening reactions of 2-azetine-2-carboxylates 3 with amines.**

**4a** 95%, 90% ee[a, f]

**4b** 93%, 91% ee[a, g]

**4c** 90%, 92% ee[a, g]

**4d** 92%, 74% ee[a, g]

**5** 96%, 99% ee[a, g]

**6** R = Bn **6a** (93%, 90% ee)[b, f]
R = 4-Br-Bn **6b** (85%, 90% ee)[b, g]

**7** 92%, 90% ee[a, g]

**8** 84%, 90% ee[a, f]

**9** 88%, 97% ee[a, f]

**10** 88% (single enantiomer)[c]

**11** 90% (single enantiomer)[c]

**12** 92%, 90% ee[a, g]

**13** Ar = Ph **13a** (65%, 90% ee)[d, g]
Ar = 4-NMe₂-C₆H₄ **13b** (93%, 90% ee)[e, g]

**14** 84% (single enantiomer)[c]

**15** 63%, 90% ee[a, g]

**16** 90% (single enantiomer)[a]

**17** 75% (single enantiomer)[b]

**18** 80%, 90% ee[b, g]

**19** 72%, 90% ee[b, g]

Reaction conditions: [a]0.2 mmol of **3**, 0.5 mmol of amine, 4.0 mL of DCM, r.t., 48 h;
[b]0.2 mmol of **3**, 0.8 mmol of amine, 4.0 mL of THF/water 1:1 (v/v), r.t., 12 h;
[c]0.2 mmol of **3**, 0.5 mmol of amine, 4.0 mL of DCM, r.t., 4 days;
[d]0.2 mmol of **3**, 0.6 mmol of amine, 4.0 mL of DCE, 65 °C, 24 h;
[e]0.2 mmol of **3**, 0.6 mmol of amine, 3.0 mL of MeNO₂, r.t., 12 h.
Enantiomeric excess determined: [f]by chiral HPLC analysis; [g]based on the ee of the original azetine

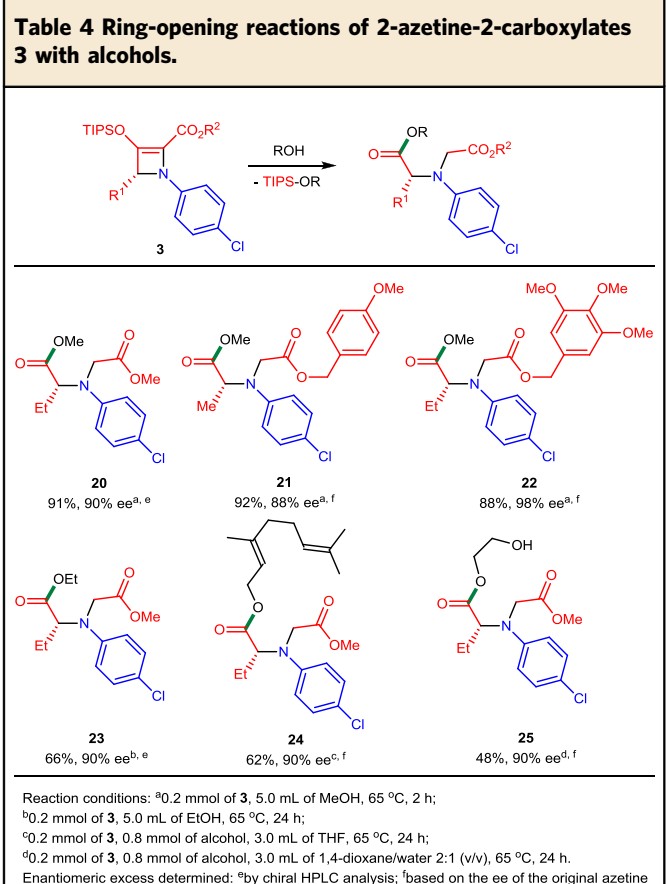

**Table 4 Ring-opening reactions of 2-azetine-2-carboxylates 3 with alcohols.**

Reaction conditions: [a]0.2 mmol of **3**, 5.0 mL of MeOH, 65 °C, 2 h;
[b]0.2 mmol of **3**, 5.0 mL of EtOH, 65 °C, 24 h;
[c]0.2 mmol of **3**, 0.8 mmol of alcohol, 3.0 mL of THF, 65 °C, 24 h;
[d]0.2 mmol of **3**, 0.8 mmol of alcohol, 3.0 mL of 1,4-dioxane/water 2:1 (v/v), 65 °C, 24 h.
Enantiomeric excess determined: [e]by chiral HPLC analysis; [f]based on the ee of the original azetine

**Table 5 Ring-opening reactions of 2-azetine-2-carboxylate 3c with other nucleophiles.**

Reaction conditions: [a]0.2 mmol of **3**, 0.3 mmol of TBAF, 4.0 mL of DCM, r.t., 2 h;
[b]0.2 mmol of **3**, 0.6 mmol of phenylhydrazine, 4.0 mL of DCE, 50 °C, 12 h;
[c]0.2 mmol of **3**, 1.0 mmol of nucleophile, 4.0 mL of THF, r.t., 12 h;
[d]0.2 mmol of **3**, 0.8 mmol of guanidine, 4.0 mL of THF/water 1:1 (v/v), r.t., 12 h;
[e]0.2 mmol of **3**, 0.6 mmol of 4-nitrophenylhydrazine, 3.0 mL of MeNO$_2$, r.t., 24 h.
[f]Enantiomeric excess reported based on the ee of the original azetine

interesting example of guanidine-based chiral amino acid **31** was obtained in the reaction of **3c** with guanidine in THF/water 1:1 (v/v). Use of 4 equiv. of guanidine produced the zwitterionic compound **31** in 78% isolated yield. The product of the attachment of two molecules of guanidine was detected in the reaction mixture by LC/MS but not isolated.

**Mechanistic studies.** That the nucleophilic ring-opening reaction carried out in DCM requires two molecules of the nucleophile is based on: (1) TIPS-Nuc was isolated as the by-product, and (2) only half of the azetine was converted to product when one equivalent of the nucleophile was used. The same outcome was observed when the reaction of **3c** was performed in THF/H$_2$O (1:1, v/v) with one equivalent of benzylamine as the nucleophile. However, complete conversion of **3c** was observed in THF within 12 h when only one equivalent each of benzylamine and water was used, which afforded ring-opened product **4a** in 92% isolated yield. We propose a sequential pathway of steps in the reaction mechanism to show all relevant reaction intermediates, including the 3-azetidiniones (Fig. 4a). The initial abstraction of the silyl (TIPS) group by a nucleophile forms 2-azetine enol **E**, the tautomer of which is 3-azetidinone carboxylate **F**. The carbonyl group of **F** then undergoes attack by the second molecule of the nucleophile to form zwitterionic four-membered ring intermediate **G** followed by ring opening to the acyclic zwitterion **H**. Rapid intramolecular proton abstraction by the carbanion forms the final amino acid derivative. We attempted to trap intermediate **H** using benzyl bromide (5 equiv.), methyl iodide (5−10 equiv.), and even dimethyl disulphide[47–51] (5 equiv.) as the electrophile, but the absence of product from substitution suggested a much higher rate for intramolecular proton transfer.

To support this mechanism, we carried out deuterium incorporation experiments on the reaction of **3c** with methanol-$d_4$ used as the solvent (Fig. 4b).

The reaction mechanism includes a set of intermediates **I**−**L** that are the same as those shown in Fig. 4a. A preparative 0.2-mmol scale ring opening reaction of **3c** with methanol-$d_4$ afforded deuterated compound **32** in 92% isolated yield in 2 h without detection of intermediates **I**−**L** by the NMR method (see Supplementary Fig. 1). To investigate a possibility of intermolecular proton transfer from intermediate **H** to the final product, we have performed a competing reaction of **3c** with benzylamine and methanol-$d_4$ (Fig. 4c). The formation of **4** as the major product confirmed the intramolecular proton transfer as the major reaction pathway. However, minor amounts of deuterated products were observed in the $^1$H NMR spectrum of the reaction mixture (see Supplementary Fig. 2) as the result of deuterium exchange on the benzylic nitrogen (~17% of D) or incorporation from methanol-$d_4$, suggesting minor competition from intermolecular proton transfer from methanol-$d_4$ during the ring opening (~33% of D). Moreover, diester **32** as the product of reaction of **3c** with benzylamine and MeOD was not observed in the reaction mixture.

To expand the scope of the ring-opened products and the synthetic applicability of the chlorine atom attached to the benzene ring, we have performed the Suzuki−Miyaura $sp^2$−$sp^2$ cross-coupling with diamide **30** (Fig. 5a). A compound containing a fluorine atom (diamide **33**) was obtained in 79% yield by treatment of diamide **30** with 4-fluorophenylboronic acid using the air and moisture stable Buchwald's third generation precatalyst [a powerful source of Pd(0)][52] that was synthesized from commercially available

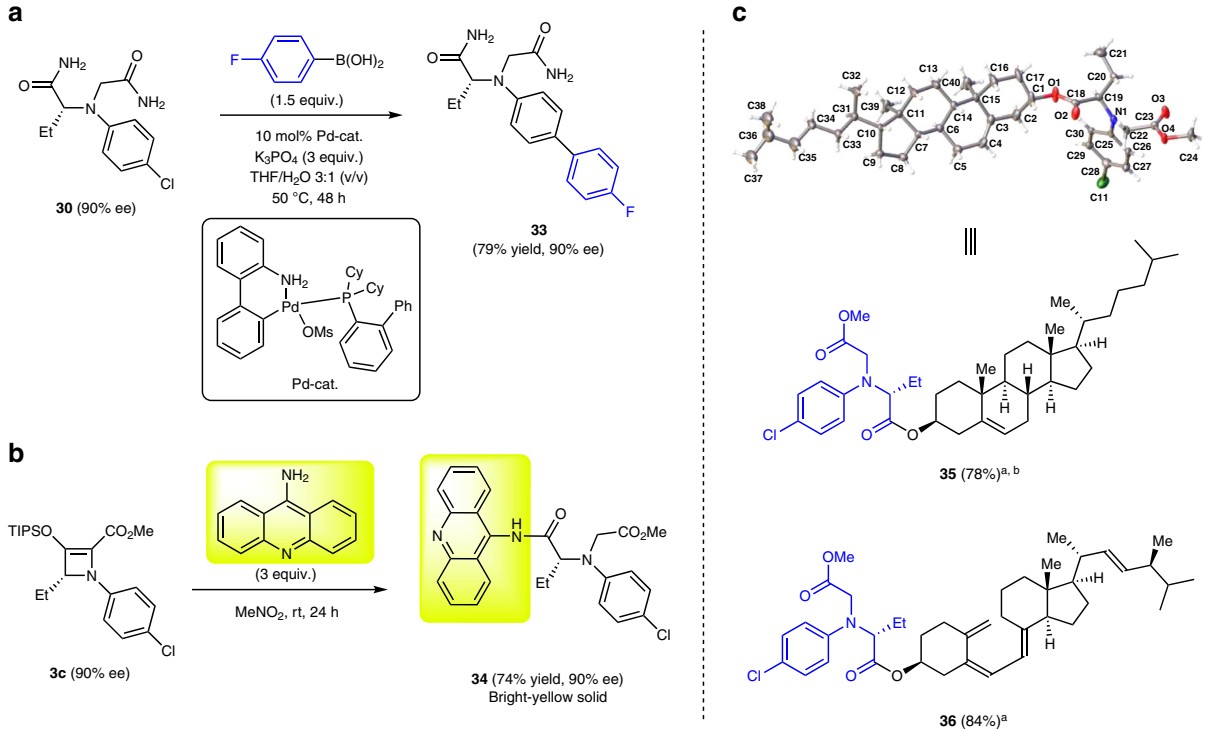

**Fig. 4** Mechanistic studies. **a** Proposed mechanism for nucleophilic ring opening of 2-azetine-2-carboxylates **3**. **b** Deuterium incorporation experiment. **c** Proton–deuterium exchange with MeOD.

**Fig. 5** Functionalization of ring-opened products. **a** Representative example of Suzuki—Miyaura cross-coupling of a ring-opened product. **b** Attachment of a fluorescent unit to chiral amino acid. **c** Functionalization of cholesterol and ergocalciferol (vitamin D$_2$) using chiral amino acid attachment.

precursors. Notably, amide functional groups remained intact under these reaction conditions; however, significant amounts of hydrolysis products were formed at temperatures over 100 °C (conversion of the amide to carboxylates).

The use of fluorophores as sensors is common in chemical biology[53,54] and plays an important role in rapid detection of peptides[55–58]. Herein, we report a robust protocol for the attachment of a fluorescent unit using the ring-opening reaction of azetine 3c with 4-aminoacridine as a fluorophore-carrying nucleophile. Bright yellow chiral amino acid derivative 34 was obtained in high yield (74%) in nitromethane as the most suitable solvent (Fig. 5b). The UV spectrum of 34 showed maximum absorptions at $\lambda = 380$, 399, and 421 nm; and the fluorescence spectrum showed maximum emissions at $\lambda = 428$, 453, and 480 nm (see Supplementary Figs. 3, 4).

As shown in Table 4, methanol and primary alcohols, but not secondary alcohols, were suitable for the ring opening of azetines. To solve the problem with secondary alcohols, we have developed a two-step protocol: synthesis of chiral monoester 26 and its reaction with naturally occurring secondary alcohols, cholesterol and ergocalciferol (vitamin $D_2$), using a classic base-catalyzed $N$, $N'$-dicyclohexylcarbodiimide (DCC) coupling reaction (Fig. 5c). Chiral diester derivatives 35 and 36 were obtained under mild conditions in 78 and 84% yields, respectively, and the structure of 35 was confirmed by X-ray crystallography establishing the absolute configuration of 26 as ($R$). This experimental evidence allowed us to assign absolute configurations of all chiral materials —azetines and ring-opened products.

## Discussion

We have developed a methodology for the strain-induced ring opening of chiral 3-azetidinone carboxylates that form glycine derivatives having high enantiopurity in high yield. The key to the success of this process is the highly enantioselective catalytic [3 + 1]-cycloaddition of silyl-protected enoldiazoacetates with azaylides that form donor–acceptor 3-triisopropylsiloxyazetine-2-carboxylates that are precursors to 3-azetidinone carboxylates. Catalysis by cationic copper(I) with a sabox chiral ligand ensures high enantiocontrol for azetine formation. Silyl group removal from the azetine by nucleophilic displacement produces the unstable 3-azetidinone-carboxylate that undergoes nucleophilic retro-Claisen ring opening. Reactions with aliphatic amines occur at room temperature, but those with the less nucleophilic arylamines and alcohols require higher temperatures. Mechanistic details, which include deuterium labeling experiments, confirm that proton transfers from the nucleophile to the azetine in the desilylation step and to the azetidinone in the retro-Claisen step are predominantly intramolecular (Fig. 4). Under specific reaction conditions, water may be the nucleophile that removes the silyl group so that the nucleophile used for the retro-Claisen ring opening can be employed in stoichiometric amounts. The rate for ring opening is dependent on the strength of the nucleophile and on the steric encumbrance around the nucleophilic center. When the reaction is performed at elevated temperatures the major competing reaction is electrocyclic ring opening of the 2-azetine carboxylate. The reaction process that is introduced provides a straightforward methodology for the functionalization of a broad selection of nucleophiles with peptide-like amine derivatives. The retro-Claisen products can be further functionalized by aryl group coupling and by esterification, suggesting broad future applications. Future efforts will be directed toward expanding the applicable nucleophiles and methods for further functionalization, as well as applying the retro-Claisen methodology to other products of catalytic asymmetric [3 + 1]-cycloaddition.

## Methods

**General procedure for [3 + 1]-cycloaddition.** To an 8-mL oven-dried screw-capped vial equipped with a magnetic stirring bar were sequentially added $Cu(MeCN)_4PF_6$ (3.72 mg, 0.0100 mmol, 5 mol%), sabox ligand L1 (8.81 mg, 0.012 mmol, 6 mol%), and 2.00 mL of dry DCM under a nitrogen atmosphere. The resulting solution was stirred at room temperature for 1 h. N-Arylsulfilimine 2a (62.2 mg, 0.200 mmol) was then introduced to the reaction solution under a flow of nitrogen, followed by dropwise addition (over 1 min) of enoldiazoacetate 1 (0.24 mmol) in dry DCM (2.00 mL). The vial was capped, and stirring was continued at room temperature for 24–72 h. Subsequently, the reaction mixture was concentrated under reduced pressure, and the residue was purified by flash chromatography on silica gel using a gradient of hexane/ethyl acetate 49:1 to 4:1 (v/v) as eluent to afford donor–acceptor azetine 3 as a pale yellow oil.

**General procedure for retro-Claisen ring opening.** To a 8-mL screw-capped vial equipped with a magnetic stirring bar an azetine 3 (0.20 mmol), an amine (0.50 mmol) and DCM (4.0 mL) were sequentially introduced. The vial was capped, and stirring was continued at room temperature for 48 h [for 4, 5, 7, 8, 9, 12, 15, 16] or 96 h [for 10, 11, 14]. After completion of the reaction (monitored by TLC), DCM was evaporated, and the residue was purified by flash chromatography on silica gel using a gradient hexane/ethyl acetate 2:1 (v/v) to pure ethyl acetate [for 4, 5, 8–12, 14] or a mixture DCM/methanol 9:1 (v/v) [for 7, 15, 16] as eluents to afford the ring opening product.

## Data availability

The authors declare that the data supporting the findings of this study are available within the paper and its Supplementary Information. Crystallographic data for compound 35 is available free of charge from the Cambridge Crystallographic Date Centre (www.ccdc.cam.ac.uk) under reference number CCDC 1910314.

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

## Acknowledgements

We acknowledge the U.S. National Science Foundation (CHE-1763168) and the Max and Minni Voelcker Fund for their support of this research. K.D. acknowledges the support from China Scholarship Council (CSC). We thank N. Greco for the synthesis of substrates and racemic samples, K. Schanze's group for the fluorescence analyses, and W.G. Griffith for mass spectral analyses. The U.S. National Science Foundation supported the acquisition of a NMR spectrometer used in this study (CHE-1625963).

## Author contributions

K.O.M. performed experiments on the synthesis of chiral azetines, their ring opening and fuctionalization, and did mechanistic studies. K.D., L.D.A., L.A.M., and K.W. performed synthesis of starting materials, racemic azetines and their ring-opened products. Y.D. performed catalyst, ligand, and sulfilimine optimizations. H.A. analyzed a sample of **35** by X-ray diffraction. K.O.M. and M.P.D. conceived and directed the project and wrote the paper. All authors discussed the results and commented on the paper.

## Competing interests

The authors declare no competing interests.
