## [Peer Review File · Nature Communications]

REVIEWERS' COMMENTS:

Reviewer #1 (Remarks to the Author):

I agree with the first reviewer on his first comment that the introduction is far too broad. The authors should concentrate on the essential with respect to the relevance of the molecules they are synthesizing. It seems to me that the motifs presented in the paper are very specific and the authors do not provide any potential applicability. What are these molecules good for? Why would someone want to use such method? These are essential questions that should be answered within the main text.

The authors state in the abstract that most methods for the asymmetric formation of a C-N or C-O bond require the use of a catalyst, or the use of stoichiometric amounts of reagents. Although this statement is true, the sequence presented in their manuscript consist of 2 non-atom economic steps, one of enantioselective catalysis for the formation of an azetine, and a second of ring opening with O and N-nucleophiles. The bond (C-N or C-O) formed through this last transformation is not involved with the stereogenic center (amide formation), the latter being created in the previous step. In consequence, the statement made in the abstract is irrelevant.

The answer of the authors to my previous comment on the use of 2 equivalents of nucleophile is rather unsatisfactory. This question could and will be raised by any reader of the manuscript, especially when it comes to using more sophisticated amines. For instance, I previously suggested that 1 equivalent of an inexpensive amine could be used, followed by the amine that the authors want to introduce. It is very disappointing that such test was not performed before resubmission.

“but the reaction scope is not limited to only 4-chlorophenyl derivatives. We did not see a substantial difference in yields and enantioselectivities with other substituents on the benzene ring.” Such additional information is appreciated in order to evaluate the potential scope of the transformation. However, if tests were performed in that regard, examples should be provided and described in the manuscript.

The reaction is not atom economic, two steps are required for the formation of motifs that have not been identified as potential pharmacological targets, starting from materials that are not readily available. Based on these facts and on previous comments, I do not think that this manuscript is suitable for the readership of Nature Communications in terms of originality, generality and relevance.

Reviewer #2 (Remarks to the Author):

This work from Doyle and co-workers represents an important addition to the preparation of highly functionalized amino acid derivatives through the use of azetines as reactive intermediates. This is enabled through the asymmetric preparation of these intermediates through a novel 3+1 cycloaddition, building a long history of underlying reactivity and catalysis in the Doyle group. This resubmission does a good job of addressing previous reviewer comments (some of which I don't agree with), and for all clearly justified reviewer points provides either a correction or an addition. While there is no question that many methods papers have more products to demonstrate scope, given the reactivity revealed herein and the scope that is reported, this reviewer feels that further extension of this chemistry is only warranted for future studies toward specialized reaction products. This work will be of great interest to the synthetic community, provides excellent yields and ee, and the SI is well done and in support of the conclusions drawn in this manuscript. Given the impact of this work, and the alterations made after the initial review, this reviewer is in full support of this manuscript.

REVIEWERS' COMMENTS:

Reviewer #1 (Remarks to the Author):

COMMENT: I agree with the first reviewer on his first comment that the introduction is far too broad. The authors should concentrate on the essential with respect to the relevance of the molecules they are synthesizing. It seems to me that the motifs presented in the paper are very specific and the authors do not provide any potential applicability. What are these molecules good for? Why would someone want to use such method? These are essential questions that should be answered within the main text.

REPLY: The Introduction has been substantially modified to clarify the methodology and its relevance. The two equations in the manuscript have been changed to Figures (Fig. 1 and 2), and the figure legends better explain the methodology. In addition, we have deleted sentences and phrases that we believed to be redundant that can be identified in Track Changes. Most importantly, the final paragraph of the Introduction has been added to summarize the major results and conclusions of this manuscript, and in doing so addresses this reviewer's concern about relevance or usefulness.

COMMENT: The authors state in the abstract that most methods for the asymmetric formation of a C-N or C-O bond require the use of a catalyst, or the use of stoichiometric amounts of reagents. Although this statement is true, the sequence presented in their manuscript consists of 2 non-atom economic steps, one of enantioselective catalysis for the formation of an azetine, and a second of ring opening with O and N-nucleophiles. The bond (C-N or C-O) formed through this last transformation is not involved with the stereogenic center (amide formation), the latter being created in the previous step. In consequence, the statement made in the abstract is irrelevant.

REPLY: The reviewer has faulted this manuscript on two issues: (1) catalyst use in the formation of

chiral azetines and (2) atom-economy in the formation of chiral azetines and in their nucleophilic ring opening. This reviewer states that our abstract gives a true statement about the use of catalysis (“this statement is true”) but then states “the statement made in the abstract is irrelevant” with claims (1) and (2) above. You will find that our manuscript gives an accurate accounting of the processes that we report without exaggeration. I do not know how to answer the criticism about atom-efficiency since this was neither a focus of this manuscript or a matter of expected judgement appropriate to this manuscript.

COMMENT: The answer of the authors to my previous comment on the use of 2 equivalents of nucleophile is rather unsatisfactory. This question could and will be raised by any reader of the manuscript, especially when it comes to using more sophisticated amines. For instance, I previously suggested that 1 equivalent of an inexpensive amine could be used, followed by the amine that the authors want to introduce. It is very disappointing that such test was not performed before resubmission.

REPLY: This is a creditable concern, although we would argue that “This question could and will be raised by any reader of the manuscript” is exaggerated and unlikely. However, we have made serious efforts in the time allotted to investigate this issue. First of all, we have undertaken the experiment suggested by this reviewer – using two different amines: primary versus secondary amines – and these 1:1 combinations give mixtures of products that would not be synthetically viable. What this reviewer failed to recognize is that the reactivity of each amine towards the silyl group and the carbonyl group in the retro-Claisen reaction is highly sensitive to both electronic and steric effects. However, our efforts did not stop with this outcome. We investigated, once again, reactions in water/THF (1:1) and discovered, as expected from our earlier studies, that the use of one equivalent of amine gave only 50% of the ring-opened product. But when only one equivalent of water was used in the THF solution, the ring opened product was formed in high yield; and this outcome is incorporated in our manuscript (in Mechanistic studies and Discussion).

COMMENT: “but the reaction scope is not limited to only 4-chlorophenyl derivatives. We did not see a substantial difference in yields and enantioselectivities with other substituents on the benzene ring.” Such additional information is appreciated in order to evaluate the potential scope of the transformation. However, if tests were performed in that regard, examples should be provided and described in the manuscript.

REPLY: We have provided this additional data. Three additional entries have been added to Table 3 (**4b**, **4c**, and **4d**), and these results are mentioned in the text just before Table 3.

COMMENT: The reaction is not atom economic, two steps are required for the formation of motifs that have not been identified as potential pharmacological targets, starting from materials that are not readily available. Based on these facts and on previous comments, I do not think that this manuscript is suitable for the readership of Nature Communications in terms of originality, generality and relevance.

REPLY: We have not claimed atom-economy. We describe an overall process that can produce chiral azetidine-2-carboxylates, store them, then apply the nucleophilic ring opening transformation. This process makes these azetidine compounds highly suitable for high-throughput screening which is valued in pharmaceutical companies.

Reviewer #2 (Remarks to the Author):

COMMENT: This work from Doyle and co-workers represents an important addition to the preparation of highly functionalized amino acid derivatives through the use of azetines as reactive intermediates. This is enabled through the asymmetric preparation of these intermediates through a novel 3+1 cycloaddition, building a long history of underlying reactivity and catalysis in the Doyle group. This resubmission does a good job of addressing previous reviewer comments (some of which I don't agree with), and for all clearly justified reviewer points provides either a correction or an addition. While there is no question that many methods papers have more products to demonstrate scope, given the reactivity revealed herein and the scope that is reported, this reviewer feels that further extension of this chemistry is only warranted for future studies toward specialized reaction products. This work will be of great interest to the synthetic community, provides excellent yields and ee, and the SI is well done and in support of the conclusions drawn in this manuscript. Given the impact of this work, and the alterations made after the initial review, this reviewer is in full support of this manuscript.

REPLY: We thank the reviewer for the thorough review of our manuscript.